# The Effect of Coupled Electroencephalography Signals in Electrooculography Signals on Sleep Staging Based on Deep Learning Methods

**DOI:** 10.3390/bioengineering10050573

**Published:** 2023-05-10

**Authors:** Hangyu Zhu, Cong Fu, Feng Shu, Huan Yu, Chen Chen, Wei Chen

**Affiliations:** 1School of Information Science and Technology, Fudan University, Shanghai 200433, China; 2Huashan Hospital, Shanghai Medical College, Fudan University, Shanghai 200040, China; 3Academy for Engineering and Technology, Fudan University, Shanghai 200433, China; 4Human Phenome Institute, Fudan University, Shanghai 201203, China

**Keywords:** deep learning, EOG, coupled EEG, sleep staging

## Abstract

The influence of the coupled electroencephalography (EEG) signal in electrooculography (EOG) on EOG-based automatic sleep staging has been ignored. Since the EOG and prefrontal EEG are collected at close range, it is not clear whether EEG couples in EOG or not, and whether or not the EOG signal can achieve good sleep staging results due to its intrinsic characteristics. In this paper, the effect of a coupled EEG signal in an EOG signal on automatic sleep staging is explored. The blind source separation algorithm was used to extract a clean prefrontal EEG signal. Then the raw EOG signal and clean prefrontal EEG signal were processed to obtain EOG signals coupled with different EEG signal contents. Afterwards, the coupled EOG signals were fed into a hierarchical neural network, including a convolutional neural network and recurrent neural network for automatic sleep staging. Finally, an exploration was performed using two public datasets and one clinical dataset. The results showed that using a coupled EOG signal could achieve an accuracy of 80.4%, 81.1%, and 78.9% for the three datasets, slightly better than the accuracy of sleep staging using the EOG signal without coupled EEG. Thus, an appropriate content of coupled EEG signal in an EOG signal improved the sleep staging results. This paper provides an experimental basis for sleep staging with EOG signals.

## 1. Introduction

Good sleep helps the body to eliminate fatigue and maintain normal brain functioning [1]. In contrast, a lack of sleep can lead to depression, obesity, coronary heart disease, and other diseases [2,3,4,5]. However, sleep disorders are becoming an alarmingly common health problem, affecting the health status of thousands of people [6,7]. To assess sleep quality and diagnose sleep disorders, signals such as electroencephalography (EEG), electrooculography (EOG), and electromyography (EMG) collected through polysomnography (PSG) are usually used to stage sleep. An entire night’s sleep can be divided into wake (W), non-rapid eye movement (NREM, S1, S2, S3, and S4), and rapid eye movement(REM) stages, according to the Rechtschaffen and Kales (R&K) standard [8] or wake (W), non-rapid eye movement (NREM, N1, N2, and N3), and rapid eye movement(REM) stages, according to the American Academy of Sleep Medicine (AASM) standard [9]. In the clinic, sleep staging is performed manually by experienced experts. The procedure is time-consuming and labor-intensive. Meanwhile, there is a subjective element in the judgment of experts and different experts do not fully agree on the classification of sleep stages [10,11]. To relieve the burden on physicians and save medical resources, many studies have focused on automatic sleep staging using biosignals through machine learning approaches [12,13,14,15].

Automatic sleep staging methods can be divided into traditional machine learning-based methods and deep learning-based methods. Traditional machine learning-based methods usually consist of handcrafted feature extraction and traditional classification methods. Handcrafted feature extraction extracts the features of signals from the time domain, frequency domain, etc., based on medical knowledge. These extracted features are then fed into traditional classifiers, such as support vector machines (SVM) [16,17,18], random forests (RF) [19,20], etc., for automatic sleep staging. Instead of requiring medical knowledge as a prerequisite, the deep learning-based method uses networks to automatically extract features. Thus, it has been widely explored in recent research [21]. Some convolutional neural networks (CNN) [22,23,24] or recurrent neural networks (RNN) [25] models have achieved good results in automatic sleep staging. Furthermore, some studies have combined different network architectures, to incorporate their advantages, such as the combination of CNN and RNN [26,27,28], the combination of RNN and RNN [29,30], and the combination of CNN and Transformer architectures [31,32]. With the development of machine learning methods, performance in sleep staging has been greatly improved, with an excellent performance on certain public datasets [33,34,35,36]. However, both the traditional machine learning-based methods and deep learning-based methods apply an EEG signal as the main or only input signal. The process of acquiring EEG signals is very tedious and uncomfortable for the subject.

Taking into account the comfort of physiological signal acquisition, some studies have tried to use certain easy-to-collect signals for sleep staging, such as cardiopulmonary signals [37,38,39], acoustic signals [40,41], and EOG signals [42,43,44]. These signals are relatively easy and comfortable to acquire compared to EEG signals, but the sleep staging performance with cardiopulmonary and acoustic signals was not satisfactory for clinical application. Noteworthy, the accuracy of sleep staging using a single-channel EOG signal was similar to that of a single-channel EEG signal in some studies [28,42]. This suggested that an EOG signal could also be used for sleep staging with good performance, allowing comfortable sleep monitoring. Despite the good results yielded by EOG signals in automatic sleep staging, the positions of the acquisition electrodes for the EOG signal and the prefrontal EEG signal are close to each other, which means that part of the EEG signal may be coupled in the EOG signal. Comparison of an EEG signal and an EOG signal in the N3 stage revealed slow wave signals with similar frequencies to those in the Fp1-O1 channel and the E1-M2 and E2-M2 channels (Figure 1). The slow wave signal, as the main characteristic wave of N3 sleep stage, appears in an EOG signal. Therefore, it is not clear whether the sleep staging ability of an EOG signal comes from the coupled EEG signal, and how this coupled EEG could affect the sleep staging results.

To explore the above issue, we conducted experiments applying data from two public datasets and one clinical dataset. First, we processed the EEG signal with a blind source separation algorithm named second-order blind identification (SOBI) [45] to obtain an EEG signal without EOG signals. Second, the raw EOG signal and the clean EEG signal were coupled to obtain a clean EOG signal and EOG signal coupled with different contents of the EEG signal. Third, the coupled EOG signals with different EEG signal contents were fed into a hierarchical neural network named two-step hierarchical neural network (THNN), which consists of a multi-scale CNN and a bidirectional gating unit (Bi-GRU), for automatic sleep staging. We also performed automatic sleep staging using the EEG signal with the THNN, to explore the difference in performance between the EOG signal and the EEG signal. Finally, we considered the impact of EEG signal coupling in EOG signals on sleep staging.

## 2. Materials and Methods

In this section, we introduce the subjects selected for this exploration, the blind source separation method used, as well as the specific structure, details, and training strategy of the THNN.

### 2.1. Subjects and PSG Recordings

In this work, we applied two widely used public datasets and one clinical dataset to conduct the experiment. The details of the three datasets are shown in Table 1.

#### 2.1.1. Montreal Archive of Sleep Studies (MASS) Dataset

The MASS dataset was provided by the University of Montreal and the Sacred Heart Hospital in Montreal [36]. It consists of whole night sleep recordings from 200 subjects aged from 18 years old to 76 years old (97 males and 103 females), divided into five subsets SS1–SS5. The SS1 and SS3 subsets have a length of 30 s for each sleep stage, the other subsets have a sleep stage of 20 s. Each epoch of the recordings in MASS was manually labeled according to the AASM standard or R&K standard by experts. The amplifier system for MASS was the Grass Model 12 or 15 from Grass Technologies. The reference electrodes were CLE or LER. In this experiment, the SS3 subset was used.

#### 2.1.2. Dreams Dataset

The DREAMS dataset was collected during the DREAMS project. It has eight subsets: subject database, patient database, artifact database, sleep spindles database, K-complex database, REM database, PLM database, and apnea database [35,46]. These recordings were annotated as microevents or as sleep stages by several experts. In this work, the subject database was applied. The subject database consists of 20 whole-night PSG recordings (16 females and 4 males) derived from healthy subjects, and the sleep stages were categorized into sleep stages according to both the R&K standard and the AASM standard. The data collection instrumentation for DREAMS was a digital 32-channel polygraph (BrainnetTM System of MEDATEC, Brussels, Belgium). The reference electrode was A1.

#### 2.1.3. Huashan Hospital Fudan University (HSFU) Dataset

The HSFU dataset is a non-public database collected in Huashan Hospital, Fudan University, Shanghai, China, during 2019–2020. Twenty-six clinical PSG recordings were collected from people who had sleep disorders. The research was approved by the Ethics Committee of Huashan Hospital (ethical permit No. 2021-811). The PSG recordings were annotated by a qualified sleep expert according to the AASM standard. The specific information of each subject is described in Table A1. The data collection instrumentation for HSFU was a COMPUMEDICS GREAL HD PSG. The reference electrodes were M1 and M2.

### 2.2. Blind Source Separation Algorithm

Blind source separation methods are widely used when dealing with coupled signals. Some common blind source separation methods include fast independent component analysis (FastICA), information maximization (Infomax), and second-order blind identification (SOBI); the first two methods require that each channel of the input signal be independent of each other, whereas SOBI has no such requirement for the input signal.Meanwhile, the effectiveness of SOBI for processing mixed signals is not affected by the number of signal channels [47]. Thus, in this experiment, the SOBI method was applied to remove interference signals, due to its robustness. The SOBI algorithm was proposed by Belouchrani et al. in 1997 [45]. This algorithm achieves blind source separation by joint approximate diagonalization of the delayed correlation matrix. It is a stable method for blind source separation. SOBI uses second-order statistics, so that it can estimate the components of the source signals with few data points. The pseudo-algorithmic of SOBI is as follows (Algorithm 1): Assuming that the input signal X has M channels and each channel has N samples, i.e., $X\in R^{M\times N}$. After normalization and whitening of the input signal, joint approximate diagonalization is performed using the covariance of the signal, to obtain the coupling coefficient. Finally, the original signal is obtained using a matrix inverse operation.
**Algorithm 1** SOBI**Input:** Input: Data, $X\in R^{M\times N}$  **Output:** Output: Data, $S\in R^{M\times N}$
  1:Normalization $X_{0}\left(t\right)\leftarrow X\left(t\right)$  2:Whitening $Z\left(t\right)\leftarrow M\left(t\right)X_{0}\left(t\right)$  3:Calculate the covariance matrix R(τ)←EZ(t+τ)Z(t)  4:**while** coefficients not converged or maximum iterations number not reached **do**  5:   Joint approximate diagonalization algorithm $U^{T}\leftarrow U^{T}R\left(\tau \right)U=I$  6:**end while**  7:Calculate the source signal $S\left(t\right)\leftarrow U^{T}Z\left(t\right)$


### 2.3. Two-Step Hierarchical Neural Network

In this work, THNN was applied to conduct automatic sleep staging. The specific structure of THNN is presented in Figure 2, and the specific parameters are shown in Table A2. THNN can be divided into two parts: the feature extraction module, and the sequence learning module. The feature extraction module uses a multi-scale convolutional neural network with two scales to extract features from different scales. The sequence learning module uses a Bi-GRU network, which can learn the temporal information in the feature matrix extracted by the feature extraction module. The feature learning module consists of a two-scale CNN network. The two scales of CNN have different sizes of convolutional kernel for extracting large-time-span features and short-time-span features in EEG signals, respectively. Specifically, if the sampling rate of an EEG signal is 128 Hz, and the convolutional kernel length of the small-scale CNN is 64, then each segment of the EEG signal is 0.5 s of the sampling signal, which corresponds to 2 Hz. The large-scale CNN has a convolutional kernel length of 640, thus each segment of the EEG signal is 5 s of the sampling signal, which corresponds to 0.2 Hz. By designing convolutional kernels of different sizes, better feature information can be extracted. Suppose the signal $S\in R^{L\times P}$ is the input of THNN, where the *L* is the number of epochs and the *P* is the length of each epoch. The process of feature learning is represented as follows:(1)$$F_{1}\;=\;scale_{1}\left(S\right)$$
(2)$$F_{2}\;=\;scale_{2}\left(S\right)$$
(3)$$F\;=\;\;{concate(F}_{1},\;F_{2})$$
where the $scale_{1}\left(\cdot \right)$ is the small scale branch of CNN, $scale_{2}\left(\cdot \right)$ is the large scale branch of CNN, and concate(·) is the concatenation layer. The sequence learning part consists of Bi-GRU, which handles the time-dependent sequence signals well and has a fast operation speed in RNN networks [48]. GRU has a fast operation speed, but it still takes a long time to train when running serially. Therefore, we added a residual structure to the serial learning module, to speed up the training [49]. Finally, the probabilities of each sleep stage were output through the softmax layer. The process of the sequence learning part is shown as follows:(4)H=GRU(F)
(5)O=residual(H,F)
(6)Y=Dense(O)
where GRU(·) is the Bi-GRU network, *H* is the temporal feature of each sleep stage outputted by the Bi-GRU, *O* is the feature after superposition of the residual module, and *Y* is the final sleep stage probability of each epoch.

### 2.4. Data Preprocessing and Experiment Scheme

In this work, we adopted the Fp1 channel EEG signal, Fp2 channel EEG signal, left EOG signal, and right EOG signal from the MASS, DREAMS, and HSFU datasets to conduct the experiments. All the signals used were filtered with a 50 Hz/60 Hz notch filter and a 0.3–35 Hz band-pass filter. and then the signals were resampled to 128 Hz to fit the network, as well as to reduce the complexity of operations. Afterwards, the SOBI method was used to remove the interference signals in the EEG signals and to obtain a clean EEG signal without the EOG signal. Next, the raw EOG signal and the clean EEG signal were processed to obtain a clean EOG signal and the EOG signal coupled with different contents of the EEG signal. The steps are showed in Figure 3. The specific calculation procedure of the coupled EOG signal is shown in Equation (Equation 7).
(7)$$coupled_{EOG}=raw_{EOG}+a*clean_{EEG}$$
where the $clean_{EEG}$ is the EEG signal without EOG signals, the $coupled_{EOG}$ is the EOG signal coupled with the EEG signal, and *a* is the superposition factor. The content of EEG signal in the EOG signal was calculated using the correlation coefficient between the coupled EOG signal and the clean EEG signal on the same side.
(8)$$correlation=corrcoef(clean_{EEG},coupled_{EOG})$$

We performed experiments using EOG signals coupled with different contents of EEG signal, and the correlation coefficients were set as 0.0, 0.1, 0.2, 0.3, and 0.5. We fed each of the five EOG signals into the network for automatic sleep staging. In addition, we used the leave-one-subject-out (LOSO) method for the validation.

## 3. Results

In this experiment, we first performed a quantitative analysis of the coupled EOG signal. Then we performed automatic sleep staging with THNN, using EOG signals coupled with different contents of EEG signals.

### 3.1. Quantitative Analysis of EOG Signals

The mean absolute error (MAE) was used to evaluate the degree of change between the coupled EOG signal and the raw EOG signal. Table 2 shows the MAE value between the raw EOG signals and coupled EOG signals. In addition, the correlation coefficients of the raw EOG signal and the clean EEG signal are presented in Table 3. It can be seen that the collected EOG signal indeed coupled with the EEG signal. The method used in this experiment allowed increasing or decrease the content of EEG signal coupled in the EOG signal. In addition, the correlation coefficient between the left eye EOG signal and the EEG signal and the correlation coefficient between the right eye EOG signal and the EEG signal were not consistent. This was particularly evident for the DREAMS dataset. In the DREAMS dataset, the correlation coefficient between the raw right eye EOG signal and the EEG signal was close to 0, demonstrating that there was almost no correlation between these two signals. This might have been due to the position of the reference electrode setting during the measurements. In general, the quantitative analysis of the EOG signal indicated that a portion of the EEG signal was indeed coupled in the EOG.

### 3.2. Sleep Staging Performance Using Coupled EOG Signals with THNN

Table 4 shows the detailed sleep staging performance of the different coupled EOG signals with THNN, including the accuracy, kappa coefficient, F1 score, specificity, and the precision of each sleep stage. With the two public datasets, the highest accuracy of automatic sleep staging using EOG signals was over 80%, and for the clinical dataset HSFU, the highest accuracy of automatic sleep staging using EOG signals was 78.9%. These results indicated that using the EOG signal for automatic sleep staging could yield good results. Meanwhile, the accuracy of automatic sleep staging using EOG signals without coupled EEG signals was also above 77% with the three datasets.

Moreover, the experimental results showed that the EEG signal coupled in the EOG signal enhanced the automatic sleep staging results, based on the EOG signal. However, an increased coupled EEG signal in the EOG signal did not allow cause improvement of the automatic sleep staging results.Specifically, in the MASS dataset, the best sleep staging results were obtained when the coupling coefficients of EOG and EEG were 0.3 (left) and 0.3 (right). In the DREAMS dataset, the sleep staging was best when the coupling coefficients were 0.3 (left) and 0.0 (right). In the HSFU dataset, the best sleep staging effect was found when the coupling coefficients were 0.3 (left) and 0.3 (right). The best sleep staging results were obtained when the amount of coupled EEG signal was moderate. Notably, these coupling coefficients were very close to the correlation coefficients of the raw EOG and EEG signals. This suggested that the raw EOG signal was a good choice for automatic sleep staging.

In addition, the classification precision of EOG signals coupled with EEG was higher for the N1, N2, and N3 stages compared to EOG signals without EEG coupling. In Figure 1b, it can be observed that the EOG signal possessed some of the recognizable waveform features of an EEG signal, such as the slow wave signal at the N3 stage. The EOG signal was coupled, to obtain a portion of the characteristic sleep waveform that would have been present only in the EEG signal, thus improving the classification accuracy of automatic sleep staging based on the EOG signal for these sleep stages. In general, the EEG signal coupled in the EOG signal was helpful for automatic sleep staging based on the EOG signal, especially for the N1, N2, and N3 stages.

Moreover, we performed sleep staging with the raw EOG signal to show the sleep staging ability of the raw EOG signal. Table 5 presents the results obtained from automatic sleep staging using the raw left and right eye EOG signals in each dataset. The results showed that the raw EOG signal also obtained a good sleep staging performance.

### 3.3. Significance Analysis

In addition, we performed significance analysis for all types of signals used in the experiments, including clean EOG signals, EOG signals coupled with different contents of EEG signal, raw EOG signals, raw EEG signals, etc. We used a chi-square test to perform a significance analysis of the sleep stage and the features extracted by the network. The specific results are shown in Table 6, where the results of the significance analysis of the coupled EOG signals are the average of the results of the EOG signals with different coupling coefficients. The results showed that the *p*-values of all the features of the signals were less than 0.05. This indicated that there was a significant relationship between the features extracted from the signals and the classification target.

## 4. Discussion

This paper investigated the question of whether the sleep staging ability of an EOG signal derives from the coupled EEG signal, and what effect the EEG signal coupled in the EOG signal has on sleep staging results. The results showed that good sleep staging results could be obtained either using EOG signals without EEG signals or coupled EOG signals. The sleep staging capability of the EOG signal came from its own characteristic information. Moreover, the accuracy of the sleep staging result using an EOG signal differed from the sleep staging results using EEG signals by only approximately 2%. This result indicated that an EOG signal can be used for automatic sleep staging with good results.

### 4.1. The Influence of Coupled EEG Signals in EOG Signals on Sleep Staging

In manual sleep scoring, the EEG coupled in EOG is usually considered noise or an interference signal. However, automatic sleep staging methods map the coupled signals from the time domain to other spatial domains through feature extraction. This allows the coupled information to be used as additional features to further complement the sleep features included in the EOG signals. In this experiment, the results showed that compared with the clean EOG signal, the EOG signal coupled with the EEG signal had a better performance for the N1, N2, and N3 stages. Note that in manual sleep scoring, the EOG signal is mainly used to classify the wake and REM stages, whereas the N1, N2, and N3 stages are commonly classified according to the EEG signal [9]. The EEG signal coupled in the EOG signal may provide features that are not in the EOG signal but are in the EEG signal, leading to enhancement of the classification results. Figure 4 shows a correlation analysis of the EOG signals for subjects in the MASS dataset with coupling coefficients of 0.0 and 0.3 for N1, N2, and N3 stages. A chi-square test was used to examine the correlation between the features extracted by the network and the classification targets under the null hypothesis that they have no correlation. In contrast, a larger chi-square value indicated a higher correlation. The results of the correlation analysis showed that *p* < 0.05, i.e., the features extracted from the EOG signals with different coupling coefficients and the classification targets were significantly correlated. Moreover, comparing the chi-square test results of the EOG signals with coupling coefficients of 0.0 and 0.3, the chi-square values of the EOG signals coupled with EEG signals were significantly higher than those of the clean EOG signals at the N1 and N2 stages. At the N3 stage, the chi-square values of the EOG signal coupled with the EEG signal and the clean EOG signal were similar. This suggested that the EOG signal coupled with the EEG signal had more effective features in the N1 and N2 stages to help the classification of sleep stage. Generally, the coupled EEG signal in an EOG signal can provide additional features that can enhance the classification accuracy of the N1, N2, and N3 stages, without affecting the classification of the wake and REM stages.

Additionally, except for the right eye EOG signal in the HSFU dataset, the best automatic sleep staging results were achieved when the MAE values between the coupled EOG signal and the raw EOG signal were the smallest, i.e., the content of the coupled EEG signal in the EOG signal was close to that in the raw EOG signal. We performed automatic sleep staging using the raw EOG signal to explore whether the raw EOG signal was sufficient to obtain a good sleep staging effect without additional EEG signal removal or addition. The results in Table 5 demonstrate that automatic sleep staging using raw EOG signals was also able to achieve good results. Fine-tuning the coupling coefficients of EEG and EOG yielded better sleep staging results, whereas this improvement was not significant compared to the results using raw EOG signals. This improvement would be lower when the sleep stages are combined for a four-class or three-class classification task. Specifically, according to the rules of the AASM [9], stages N1 and N2 can be combined as light sleep stages and N3 can be considered a deep sleep stage, thus becoming a four-class task (W, light sleep, deep sleep, REM). Therefore, when using the EOG signal for automatic sleep staging, the use of the raw EOG signal can yield sufficient sleep staging results. Further fine-tuning of the EEG and EOG coupling coefficients did not significantly improve the accuracy of the automatic sleep staging results.

### 4.2. The Difference of the Sleep Staging Results Using EEG Signal and EOG Signal

The experiment results showed that the difference between automatic sleep staging using coupled EOG signals and EEG signals was not significant. However, the classification accuracy at the N2, N3, and REM stages between using EOG and EEG signals was significant. Figure 5 shows a comparison of the coupled EOG signal with a 0.3 coefficient and a raw EEG signal of a subject at the N2, N3, and REM stages in the MASS dataset. At the N2 stage, the amplitude of the EOG signal was lower compared to the EEG signal, which led to the values of the extracted features being smaller, resulting in a decrease in the classification accuracy. For the frequency domain, the energy of the EOG signal was also lower than that of the EEG signal. At the N3 stage, the waveforms of the EOG and EEG signals were relatively similar. In the frequency domain, the EOG signal covered a much lower frequency range and provided less information. In the REM stage, the EOG signal produced large amplitude changes in a short period of time, which is characteristic of the REM stage eye movements. These eye movements greatly enhanced the classification accuracy of the EOG signal in the REM stage. In the frequency domain, the spectral energy of the EOG signal was also much larger than that of the EEG signal, which contributed to the high classification accuracy of the EOG signal in the REM stage. Overall, the EOG signal and the EEG signal both have advantages for sleep staging. The EOG signal contains features that make it better at classifying the wake and REM stages. The EEG signal is better at classifying the N1, N2, and N3 sleep stages. The EOG signal coupled with the partial EEG signal also obtained part of the sleep staging characteristics of the EEG signal, which provides a good basis for comfortable sleep staging, as well as sleep monitoring using the EOG signal.

### 4.3. Limitations and Future Works

Based on this experiment, we need to point out some limitations. First, we explored sleep staging with the EOG signal in normal subjects and in subjects with sleep disorders. However, these subjects were all adults. In future work, experimental exploration of differently aged people could be attempted to extend the applicability of the findings of this paper. Second, the blind source separation method we used in this work was not the most advanced, and there are other methods such as artifact subspace reconstruction (ASR), morphological component analysis (MCA), and surrogate-based artifact removal (SuBAR). These methods could be tried in future works, to remove interference signals. Third, we only used one network for sleep analysis of the EOG signals. There are many better network models available, such as XSleepNet [30], TransSleep [50], AttnSleep [31], etc. In future work, better networks could be tried, to improve the sleep staging results of EOG signals. Fourth, in terms of signal acquisition, EMG signals are also easy to acquire. Meanwhile, the submental EMG signal is of great significance for the differential staging of wake and REM stages (being especially important for the differential diagnosis of REM sleep behavior disorder) [51]. In future work, we could try to use both EOG and EMG modalities as inputs to the network, to explore a portable sleep monitoring method with a better staging effect. Finally, EOG signals and EEG signals are complementary to each other for sleep staging. They should be combined rather than separated. Therefore, in future work, we will work on combining EEG signals and EOG signals, to improve automatic sleep staging and to investigate comfortable and efficient sleep monitoring, based on the fact that both prefrontal EEG and EOG signals are easy to acquire.

## 5. Conclusions

In this paper, we investigated the effect of the EEG signal coupled in an EOG signal on sleep staging results. Two publicly available datasets and one clinical dataset were used for the experiment. The SOBI method was applied to obtain a clean EEG signal. The clean EEG signal and the raw EOG signal were used to obtain a clean EOG signal and EOG signal coupled with different contents of the EEG signal. Afterwards, a THNN was used to perform automatic sleep staging with coupled EOG signals. The results showed that the sleep staging capability of the EOG signal was not derived from the coupled EEG signal but from its own feature information. Meanwhile, the EOG signal coupled with the EEG signal had better classification performance for the N1, N2, and N3 stages. The coupled EEG signal could complement the feature information lacking in the EOG signal, especially in the N1 and N2 stages. In addition, the amount of coupled EEG signal was similar to the amount of EEG signal contained in the raw EOG signal. Higher or lower levels than this could result in a certain reduction in the accuracy of the sleep staging results. This paper provided an explorative experimental analysis of automatic sleep staging using EOG signals. Moreover, it is excepted to provide an experimental basis for comfortable sleep analysis, home sleep monitoring, etc.

## Figures and Tables

**Figure 1 bioengineering-10-00573-f001:**
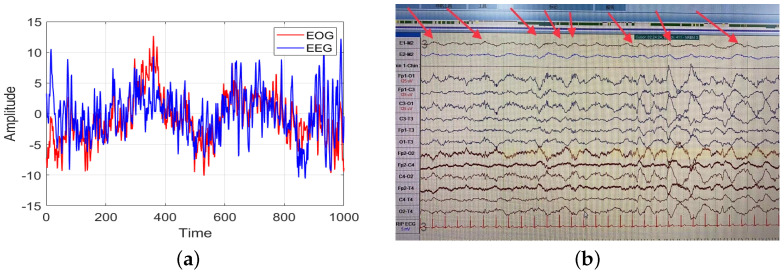
Comparison of EOG signal and prefrontal EEG signal. (**a**) Comparison of EOG signal and EEG signal waveforms within one epoch. (**b**) One epoch of a PSG signal at stage N3, including EOG, EEG, and ECG signals.

**Figure 2 bioengineering-10-00573-f002:**
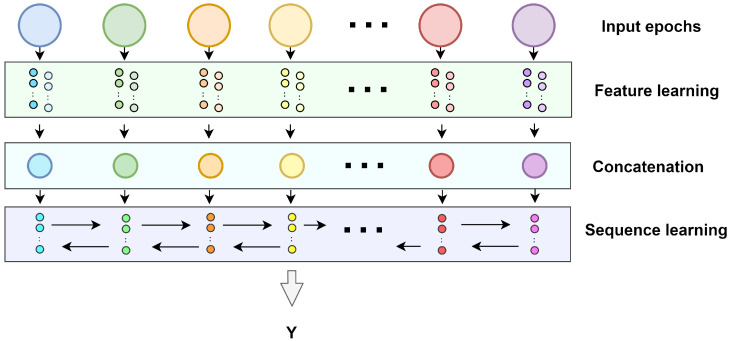
The structure of THNN. The input signal is first extracted using features at different scales by the feature extraction part. The features are concatenated and sent to the sequence learning module for further extraction of the temporal features in the signal.

**Figure 3 bioengineering-10-00573-f003:**
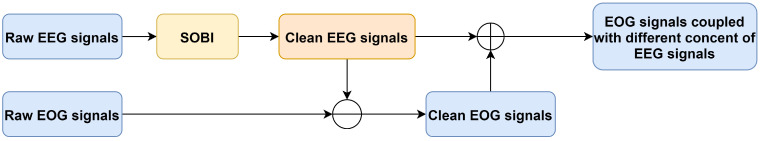
The process of obtaining EOG signals coupled with different contents of EEG signal.

**Figure 4 bioengineering-10-00573-f004:**
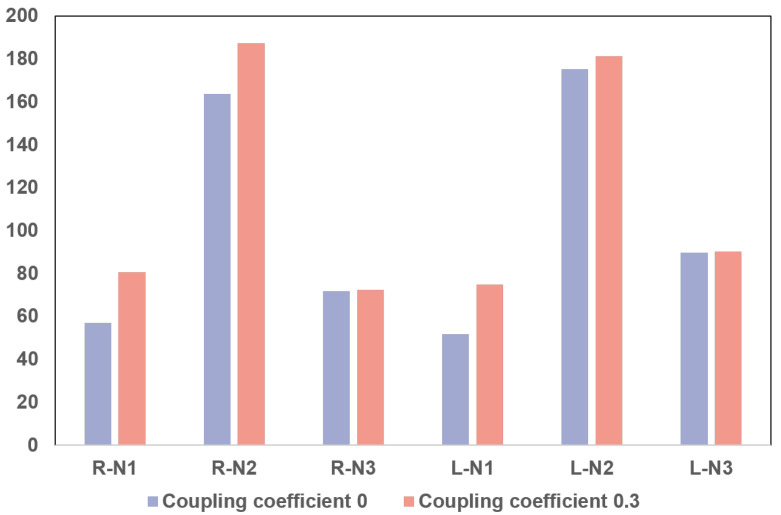
Results of the chi-square test for EOG signals with different coupling coefficients in the MASS dataset.

**Figure 5 bioengineering-10-00573-f005:**
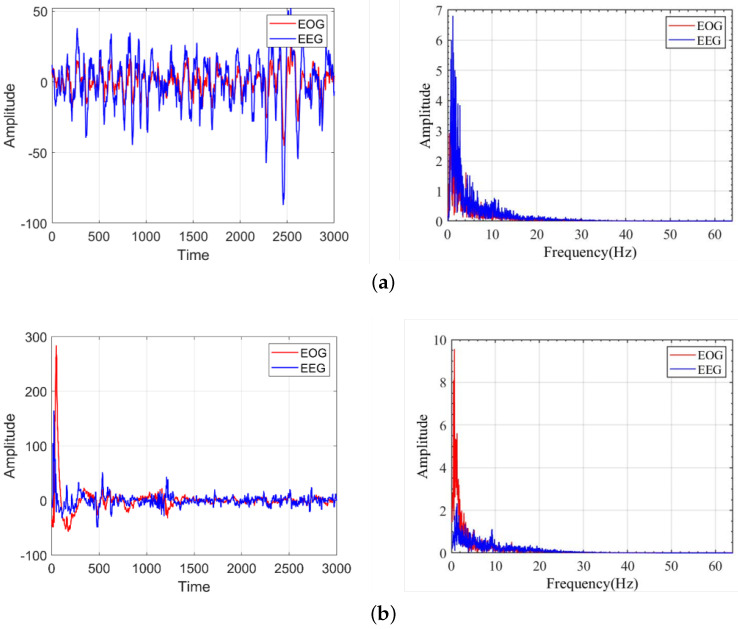
Comparison of time and frequency domains of a coupled EOG signal with a 0.3 coupling coefficient and an EEG signal in the N2 and REM stages in the MASS dataset. (**a**) Comparison of the time domain and frequency domains of the EOG signal with the best results and EEG signal in the N2 stage in MASS dataset. (**b**) Comparison of time domain and frequency domains of the EOG signal with the best results and the EEG signal in the REM stage for the MASS dataset.

**Table 1 bioengineering-10-00573-t001:** Details of the MASS, DREAMS, and HSFU datasets.

Dataset	Subjects	Age	Healthy	Sampling Rate	Stages					
					W	N1	N2	N3	REM	Total
DREAMS	20	20–65	Yes	200 Hz	3551	1480	8251	3933	3019	20,234
MASS	62	18–76	Yes	256 Hz	6442	4839	29,802	7653	10,581	59,317
HSFU	26	25–65	No	1024 Hz	7278	2926	9507	3001	4082	27,194

**Table 2 bioengineering-10-00573-t002:** MAE value (uV) between the raw EOG signal and coupled EOG signal.

Dataset	MASS		DREAMS		HSFU	
**Correlation Coefficient**	**L**	**R**	**L**	**R**	**L**	**R**
0.0	5.00 ± 1.24	2.98 ± 1.26	0.21 ± 0.08	0.01 ± 0.01	0.27 ± 0.06	0.32 ± 0.08
0.1	3.86 ± 1.19	2.18 ± 0.91	0.15 ± 0.08	0.02 ± 0.01	0.21 ± 0.06	0.26 ± 0.08
0.2	2.72 ± 1.21	1.42 ± 0.82	0.11 ± 0.07	0.04 ± 0.02	0.15 ± 0.06	0.20 ± 0.08
0.3	1.70 ± 1.13	0.98 ± 0.98	0.08 ± 0.06	0.06 ± 0.03	0.08 ± 0.06	0.13 ± 0.09
0.5	1.70 ± 1.26	2.59 ± 1.75	0.13 ± 0.09	0.11 ± 0.06	0.09 ± 0.06	0.08 ± 0.04

L, left eye; R, right eye.

**Table 3 bioengineering-10-00573-t003:** Correlation coefficient between the raw EOG and clean EEG signal.

Dataset	MASS		DREAMS		HSFU	
**Right or Left Eye**	**L**	**R**	**L**	**R**	**L**	**R**
Correlation coefficient	0.43 ± 0.11	0.30 ± 0.23	0.32 ± 0.12	0.01 ± 0.04	0.40 ± 0.08	0.46 ± 0.10

L, left eye; R, right eye.

**Table 4 bioengineering-10-00573-t004:** Results of the MASS, DREAMS, and HSFU datasets with THNN.

Dataset		Cor	Acc	Kappa	B	Spec	Precision of Each Stage				
							Wake	N1	N2	N3	REM
MASS	Right EOG	0.0	0.790	0.786	0.691	0.885	0.654	0.308	0.825	0.797	0.841
		0.1	0.790	0.786	0.689	0.880	0.626	0.353	0.848	0.811	0.847
		0.2	0.799	0.795	0.700	0.889	0.674	0.361	0.868	0.805	0.848
		0.3	0.804	0.796	0.688	0.883	0.654	0.397	0.848	0.862	0.785
		0.5	0.765	0.759	0.669	0.866	0.605	0.391	0.823	0.748	0.831
	Left EOG	0.0	0.791	0.786	0.699	0.885	0.629	0.303	0.823	0.792	0.851
		0.1	0.792	0.787	0.689	0.883	0.681	0.327	0.845	0.803	0.866
		0.2	0.803	0.799	0.696	0.884	0.697	0.343	0.839	0.826	0.880
		0.3	0.803	0.798	0.708	0.890	0.690	0.358	0.868	0.801	0.839
		0.5	0.762	0.757	0.664	0.865	0.623	0.339	0.819	0.742	0.826
	EEG	-	0.817	0.806	0.688	0.898	0.665	0.398	0.901	0.741	0.794
DREAMS	Right EOG	0.0	0.811	0.808	0.749	0.896	0.882	0.518	0.794	0.817	0.856
		0.1	0.809	0.806	0.749	0.895	0.891	0.578	0.802	0.738	0.744
		0.2	0.807	0.803	0.743	0.893	0.874	0.526	0.798	0.777	0.883
		0.3	0.797	0.793	0.736	0.888	0.869	0.524	0.784	0.784	0.850
		0.5	0.791	0.787	0.725	0.885	0.826	0.498	0.784	0.793	0.839
	Left EOG	0.0	0.806	0.803	0.759	0.892	0.880	0.586	0.784	0.779	0.905
		0.1	0.800	0.796	0.742	0.889	0.870	0.560	0.786	0.757	0.891
		0.2	0.808	0.805	0.750	0.894	0.875	0.587	0.795	0.760	0.899
		0.3	0.811	0.807	0.748	0.895	0.896	0.605	0.765	0.792	0.824
		0.5	0.809	0.806	0.747	0.896	0.886	0.595	0.810	0.749	0.859
	EEG	-	0.820	0.816	0.743	0.903	0.887	0.599	0.840	0.811	0.756
HSFU	Right EOG	0.0	0.774	0.766	0.683	0.872	0.916	0.163	0.730	0.702	0.870
		0.1	0.778	0.770	0.682	0.878	0.934	0.227	0.763	0.680	0.867
		0.2	0.785	0.778	0.676	0.878	0.929	0.275	0.746	0.752	0.823
		0.3	0.789	0.782	0.676	0.879	0.928	0.268	0.741	0.751	0.851
		0.5	0.782	0.774	0.658	0.879	0.936	0.163	0.774	0.724	0.761
	Left EOG	0.0	0.774	0.764	0.654	0.878	0.934	0.136	0.761	0.645	0.800
		0.1	0.777	0.769	0.676	0.880	0.939	0.246	0.790	0.633	0.843
		0.2	0.788	0.780	0.684	0.884	0.925	0.252	0.790	0.656	0.867
		0.3	0.789	0.781	0.683	0.881	0.941	0.333	0.782	0.719	0.816
		0.5	0.768	0.759	0.657	0.873	0.900	0.229	0.781	0.665	0.786
	EEG	-	0.791	0.781	0.656	0.884	0.859	0.263	0.802	0.841	0.703

Cor, correlation; Acc, accuracy; Spec, specificity; EOG, electrooculography; EEG, electroencephalography.

**Table 5 bioengineering-10-00573-t005:** Results of sleep staging using raw EOG signals.

	Left EOG			Right EOG		
** Dataset**	**Accuracy**	**Kappa**	**F1-Score**	**Accuracy**	**Kappa**	**F1-Score**
MASS	0.799	0.793	0.705	0.800	0.793	0.705
DREAMS	0.809	0.806	0.749	0.807	0.803	0.739
HSFU	0.788	0.779	0.666	0.787	0.780	0.689

**Table 6 bioengineering-10-00573-t006:** The *p*-value of the chi-square test for different signals in different datasets.

	L		R		
** Dataset**	**Raw EOG**	**Coupled EOG**	**Raw EOG**	**Coupled EOG**	**Raw EEG**
MASS	0.011	0.010	0.010	0.010	0.006
DREAMS	0.015	0.013	0.014	0.013	0.009
HSFU	0.023	0.020	0.022	0.021	0.019

L, left eye; R, right eye.

## Data Availability

MASS dataset is at http://www.ceams-carsm.ca/en/MASS (accessed on 8 June 2020), DREAMS dataset is at http://www.tcts.fpms.ac.be/%7Edevuyst/#Databases (accessed on 8 June 2020) and the HSFU dataset is not available.

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
