# Peer review of "The Effect of Coupled Electroencephalography Signals in Electrooculography Signals on Sleep Staging Based on Deep Learning Methods"

_bioengineering, 2023, doi:10.3390/bioengineering10050573_

Round 1

Reviewer 1 Report

This interesting article investigated possibility of improving automatic sleep staging by using electrooculography and deep mashine learning.

I find the paper scientifically and clinically important and sound so I suggest it to be accepted.

My only objection is to remove all tables and figures from disscussion into results section

Reviewer 2 Report

The Authors compare the effectiveness of electrooculography and electroencephalographic signals in automatic sleep staging based on deep learning strategies.

The approach of the manuscript seems to be quite promising, but some minor modifications have to be implemented by the authors in order to clarify their approach and to assess the effective contribution of their paper.

1)  In line 136, the reference to the frequency unit does not seem justified, as signal processing is only realized in the time domain. The impact of the time window on frequency resolution could be appreciated if a time-domain to frequency-domain signal transformation were applied.

2)    The instrumentation used in the three databases considered should be reported (In particular, the locations of Reference and Ground electrodes)

3)    The correlation coefficient detects a coupling of the two signals downstream of a mixing. It is not clear how the mixing takes place and how it is modulated to obtain the desired values of the stated correlation.

4)    There is a lack of clarification as to how the classification of different sleeping stages is realized, i.e., within a multi-class problem or several binary problems.

Minor concerns:

-       The reference of the second-order blind identification method (SOBI) should be given the first time it is mentioned.

-       Table I: the label “Stages” should be collocated over the sleep stages and substituted by the label “total”

-       Table 2: Please insert in the caption the meaning of  “L” and “R”

Reviewer 3 Report

  1. The authors report the effect of coupled EEG signals in EOG signals on sleep staging based on deep learning methods, explored in 2 public datasets and 1 sleep clinic dataset, in total 108 PSGs. The accuracy of the coupled EEG signal was around 80% in all 3 datasets without coupled EEG signals. The authors conclude that appropriate content of coupled EEG signal in EOG signal could improve the sleep staging results. The topic is important and the results may contribute to a more time-saving approach in sleep staging and management of the patients with sleep disorders. However, there are some concerns that should be answered in order to improve the manuscript.

    1) In the Materials and Methods section, the number of PSGs in the MASS dataset is 200 but in Table 1, it's 62. Please clarify.

    2) There's no subsection, "Statistics". Please add and clarify.

    3) EOG signal is particularly important for REM sleep, and the submental EMG signal is of high interest in differential staging wake vs REM sleep (particular clinical importance for the differential diagnosis of REM sleep behavior disorder). How the accuracy values be if the authors considered the submental EMG signals in comparisons; i.e., EOG without EEG in combination with submental EMG?

    4) The authors state that the interference in raw EEG signals was removed using SOBI; however no detail is given regarding how the raw EEG and raw EOG were processed to obtain clean EOG signal. Was blind source separation used again?

    5) Similarly, how was the coupled EOG signal produced using clean EEG and EOG signals to ensure different correlation values?

    6) Which unit does the MAE leverage in comparing raw EOG and coupled EOG? Is it uV?
